# A *TSHZ3* Frame-Shift Variant Causes Neurodevelopmental and Renal Disorder Consistent with Previously Described Proximal Chromosome 19q13.11 Deletion Syndrome

**DOI:** 10.3390/genes13122191

**Published:** 2022-11-23

**Authors:** René G. Feichtinger, Martin Preisel, Katja Steinbrücker, Karin Brugger, Alexandra Radda, Saskia B. Wortmann, Johannes A. Mayr

**Affiliations:** 1Department of Pediatrics, Salzburger Landeskliniken (SALK) and Paracelsus Medical University (PMU), 5020 Salzburg, Austria; 2Department of Pediatrics, Hospital Villach, 9500 Villach, Austria; 3Amalia Children’s Hospital, Radboudumc, 6525 GA Nijmegen, The Netherlands

**Keywords:** TSHZ3, neurodevelopmental disorder, renal disorder, novel disease loci

## Abstract

Heterozygous deletions at 19q12–q13.11 affecting TSHZ3, the teashirt zinc finger homeobox 3, have been associated with intellectual disability and behavioural issues, congenital anomalies of the kidney and urinary tract (CAKUT), and postnatal growth retardation in humans and mice. *TSHZ3* encodes a transcription factor regulating the development of neurons but is ubiquitously expressed. Using exome sequencing, we identified a heterozygous frameshift variant c.119_120dup p.Pro41SerfsTer79 in *TSHZ3* in a 7-year-old girl with intellectual disability, behavioural issues, pyelocaliceal dilatation, and mild urethral stenosis. The variant was present on the paternal TSHZ3 allele. The DNA from the father was not available for testing. This is the first report of a heterozygous point mutation in *TSHZ3* causing the same phenotype as reported for monoallelic deletions in the same region. This confirms *TSHZ3* as a novel disease gene for neurodevelopmental disorder in combination with behavioural issues and CAKUT.

## 1. Introduction

*TSHZ3* encodes teashirt zinc finger homeobox 3, a 118.6 kDa protein (https://www.uniprot.org, accessed on 1 August 2022) containing five zinc fingers important for its function in transcriptional regulation. It is highly expressed in the fetal neocortex and in peri-urothelial cells of the proximal ureter and renal pelvis at 9 weeks of gestation [1,2].

The TSHZ3 gene consists of two exons and resides within a region 19q12 proximal to chromosomal region 19q13.11. Heterozygous deletions in this region (chromosome 19q13.11 deletion syndrome; proximal; MIM #617219) causing TSHZ3 haploinsufficiency have been related to a neurodevelopmental disorder (NDD) characterized by developmental delay, absent or delayed speech, intellectual disability, and behavioural issues/autistic features. Some individuals show congenital anomalies of the kidney and urinary tract (CAKUT)**.** Furthermore, dysmorphic features, tapered fingers, fifth finger clinodactyly, clubfoot, feeding difficulties, pyloric stenosis, hip dysplasia, and muscular hypotonia were reported in some of the individuals [1].

In 1998, one individual with a deletion encompassing several genes in 19q13.11 including *TSHZ3* was described [3]. Later, 13 individuals with 19q13.11 deletions including the TSHZ3 locus and one individual in whom only exon 2 and parts of the intron of *TSHZ3* were missing were reported [1,4,5,6]. This adds to a growing pool of associations between TSHZ3 and an NDD with behavioural issues with/without CAKUT. However, *TSHZ3* is currently not listed as a disease gene in OMIM.

*TSHZ3* is thought to be a transcriptional regulator of the development of cortical projection neurons involved in both respiratory rhythm and airflow control and the differentiation of the proximal uretic smooth muscle cells [1]. Homozygous knock out mice failed to take their first breath, quickly became cyanotic, and died. In addition, one half of the heterozygous knock out mice died shortly after birth. The survivors showed an initial growth retardation but soon reached normal length [1]. Mice with *Tshz3* haploinsufficiency show autism spectrum disorder-like behaviour (social interaction deficits, restricted fields of interest, and stereotypical autistic behaviour) [7]. It was also reported that the particular cortical gene expression of genes encoding for glutamatergic synapse components were altered [8].

## 2. Materials and Methods

We performed whole-exome sequencing from leucocyte-derived DNA. The library was prepared by SureSelect60Mbv6 (Agilent) and was paired-end sequenced on a HiSeq 4000 platform (Illumina) with a read-length of 100 bases [9]. Reads were aligned to the human genome assembly hg19 Burrows–Wheeler Aligner (BWA, v.0.5.87.5) and detection of genetic variation was performed using PINDEL (v 0.2.4t) and ExomeDepth (v 1.0.0). The cut-off for biallelic inheritance was set to <1% allele frequency and at <0.1% for monoallelic inheritance. The size of reference entries constituted >27,000 exomes in the in-house database at the time of analysis. A total of 97.1% of all regions had > 20× coverage. A total of 78832 SNV were detected and 11,007 indels. After filtering, we had 112 rare variants (allelic frequency less than 6 in 27,503 in our database) and 100 variants with two hits in one gene (either bi-allelic or mono-allelic, with an allelic frequency less than 551 in 27,503 in our database). Reads were aligned to human genome build GRCh37/hg19 and assessed for sequence alterations using a custom-made bioinformatics tool [9]. Sanger sequencing was performed using standard methods. The participant and her parents provided written and informed consent for the taking and distribution of the photographs. The forms are stored in compliance with local confidentiality laws.

## 3. Results

### 3.1. Clinical Description

This girl was the only child of unrelated Caucasian parents. She was born by Caesarean section (week of gestation 38 + 3) after an uneventful pregnancy and had an unremarkable adaptation (APGAR 9/10/10 at 1/3/5 min, birth weight 2540 g (p5), length 48 cm (p13), and head circumference 33 cm (p14)).

She had a febrile urinary tract infection at the age of 2 and 4 months. An initial ultrasound at 2 months of age showed a marked pelvic region of the right kidney, while a follow-up ultrasound at the age of 4 years and 7 months showed grade 2 pyelocaliceal dilatation.

The girl was slightly delayed in achieving early motor milestones (e.g., walking independently from the age of 24 months onwards; she was unable to ride a bike but could to swim by the age of 6.5 years), but mainly showed a delay in her receptive and expressive speech and linguistic, intellectual, and social-emotional development.

At the age of 6.9 years, her physical and neurological examination was unremarkable with the exception of being diagnosed as overweight (weight 37 kg (*p* > 99), possessing a height of 124 cm (p.68)), and presenting a bilateral intention tremor as well as a spontaneous tremor of the right first toe (Figure 1C). She showed behavioural problems, which were difficult to classify. The observations of her mother using the Child Behaviour Checklist (CBCL/6-18R) were identical with the observation by the medical team. The girl showed difficulties in both internal as well as external areas (anxious/depressed, social problems, attention problems, rule-breaking behaviour, and aggressive behaviour).

At the biological age of 6.9 years, she was extensively tested.

The Wechsler Intelligence Scale for Children 5th edition (WISC-V) showed a borderline total IQ of 77 (mean 100; SD *15)*, percentage rank (PR) 6), with the following subscales: Verbal Comprehension Index IQ—100/PR 50; Fluid Reasoning Index IQ—86/PR 18, Working Memory Index IQ—79/PR8; Processing Speed Index IQ—80/PR9).

Her speech and language development were tested using German test batteries (e.g., TROG-D for grammar comprehension and PDSS for vocabulary, the *Vienna test for sentence completion*, and *Scenotest* (a general developmental projective test in the German language)) and showed deficits in all areas with a speech age of about 4–5 years.

She had major problems with motor skills (fine motor skills, hand coordination, body coordination, strength, and skills, all tested with the Bruininks–Oseretzky test; graphomotoric skills were tested with the house-tree-person drawing test)), wherein she showed a developmental age of 4–4.9 years.

In general, it was noticed that she had many problems with everyday tasks, e.g., she was unable to put her socks on or dress herself. Her overweight was attributed to a sensory deficit in body perception. Furthermore, she had behavioural problems. The Child Behavior Checklist (CBCL/6-18R) showed above average external (T = 83) and internal (T = 65) problems and a pathological total value (T = 79). Taken together, her developmental age was 4.0–4.9 years at the biological age of 6.9 years.

### 3.2. Genetic Testing

An exome analysis revealed a monoallelic frameshift variant in *TSHZ3* (NM_020856.4) c.[119_120dup];[=] (p.[Pro41SerfsTer79];[=]), which was absent from gnomAD [pLI = 0.99, o/e = 0.11 (0.05–0.28)] (Figure 1A, Appendix A). The exome-sequencing depth at the *TSHZ3* variant c.119_120dup was 224 (52% of these were duplications). The TSHZ3 frameshift was not detectable in the DNA of the mother. Unfortunately, no DNA of the father could be obtained. Since the maternally inherited TSHZ3 allele contains two polymorphisms close to the frame-shift mutation, we used a restriction digest of the patient’s DNA in order to discriminate the two TSHZ3 alleles. After digestion, PCR amplification, and Sanger sequencing, we could detect that the TSHZ3 variant was located on the paternal allele (Appendix A). According to the ACMG guidelines, the variant is classified as pathogenic. *TSHZ3* contains only two exons: a very short first one (encoding 14 amino acids) and a very large one (encoding 1067 amino acids). This very early termination should, therefore, cause a complete loss of protein function or haploinsufficiency since the resulting protein shows a lack of all five functional zinc fingers (Figure 1 B). In gnomAD, six LoF variants are listed: c.41-1G>T; p.Glu66Ter; p.Gln177Ter; p.Tyr279Ter; p.Met1028AsnfsTer30; p.Pro1068ArgfsTer11; p.Leu1072ProfsTer6. Since affected individuals are also included in gnomAD, we examined the gnomAD v.2.1.1 (controls) data set only containing healthy individuals. Here, only the p.Met1028AsnfsTer30 and p.Leu1072ProfsTer6 LoF variants were found. The p.Leu1072ProfsTer6 variant is flagged in gnomAD for low confidence (LCpLoF). Only the p.Met1028AsnfsTer30 variant remains in the gnomAD control population. However, this mutation, in contrast to the described mutation, affects the outermost C-terminus of TSHZ3 and, therefore, might allow for the production of a stable transcript with some residual function.

## 4. Discussion

Numerous microdeletion disorders were described prior to the identification of the specific disease genes residing within the critical deletion region. Initially, 9q deletions were associated with Kleefstra syndrome (9q Subtelomeric Deletion Syndrome; 9q34.3 Microdeletion Syndrome; 9qSTDS) (https://www.ncbi.nlm.nih.gov/books/NBK47079/, accessed on 1 August 2022); later, it was recognized that single nucleotide variants (SNVs) in the *EHMT1* gene also cause Kleefstra syndrome (MIM#607001). Further prominent examples include Van-Asperen syndrome/chromosome 17q11.2 deletion syndrome (MIM#613675) harbouring *NF1* (MIM#613113), Koolen-De Vries syndrome harbouring *KANSL1* (MIM#610443), Miller–Dieker syndrome (MIM#247200) harbouring *PAFAH1B1* (MIM#601545), and Smith–Magenis Syndrome/17p11.2 microdeletion syndrome including *RAI1* (MIM#182290).

Herein, we provide further evidence that *TSHZ3* is the responsible disease gene associated with the phenotype observed in chromosome 19q13.11 deletion syndrome, which is proximal (MIM #617219) in both human and knock out mouse models. The individual presented here is the first reported with a monoallelic SNV in *TSHZ3* associated with an NDD with behavioural features and CAKUT. In the DECIPHER database (https://www.deciphergenomics.org/, accessed on 1 August 2022), a patient carrying a de novo LoF variant in TSHZ3 c.[366C>A];[=]; p.[Tyr122Ter];[=] and abnormalities of the nervous system is described. The involvement of the nervous system would fit with our individual, although no information concerning the kidneys are given in DECIPHER.

A total of 14 individuals with heterozygous deletions encompassing TSHZ3 including one with an intragenic deletion (patient 7 from Caubit et al.) have been described in the literature so far [1]. In eight of these, the copy number variants were not detectable in the parental leucocyte-derived DNA and presumably occurred in a heterozygous manner. In another, three of these were inherited from the (healthy) parent, suggesting incomplete penetrance. In three cases, the mode of inheritance was not available. A heterozygous missense variant c.[724G>C];[=] p.[Asp242His];[=] was described by Nicolaou et al. in a series of CAKUT patients. Although the renal phenotype would fit with our study, it is doubtful that the heterozygous missense variant has pathogenic relevance [10]. According to the ACMG classification, this variant is benign (BP1; BP4; PM2).

The phenotype of our individual (NDD, behavioural issues, and grade 2 pyelocaliceal dilatation) is in line with the previously reported cases (intellectual disability (12/14), speech delay (13/14), behavioural issues including autistic features (8/8), and renal tract abnormalities 8/8) and the mouse model. Our individual also showed postnatal growth retardation, which was previously reported in humans (9/14) and mice, but no neonatal feeding difficulties that were described frequently in other TSHZ3 haploinsufficient cases (12/14).

In summary, heterozygous CNVs encompassing TSHZ3, intragenic deletions, and specific monoallelic SNVs in *TSHZ3* might be associated with an NDD with behavioural/autistic features with/without CAKUT. RNA sequencing should be performed to enforce the description of this new variation. Therefore, we propose *TSHZ3* as the causative disease gene. The combination of neurodevelopmental issues and (mild) CAKUT can aid variant interpretation.

## Figures and Tables

**Figure 1 genes-13-02191-f001:**
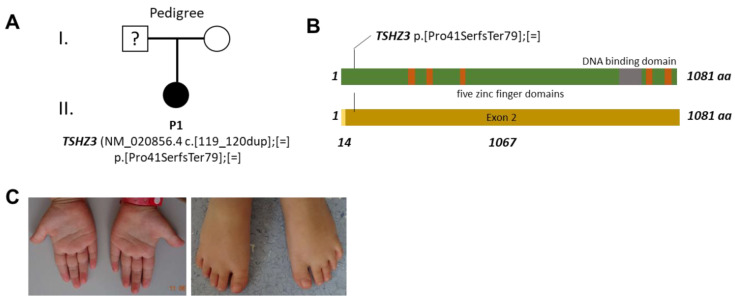
Pedigree, TSHZ3 domain architecture, variant position, and habitus of the affected individual. (**A**) Pedigree of the family of the affected individual. (**B**) TSHZ3 domain architecture. DNA binding domain = grey; zinc fingers = brown. (**C**) Clinical photos of the hands and feet of the affected individual at the age of 6.9 years. The participant and her parents provided written and informed consent for the taking and distribution of the photographs. The forms are stored in compliance with local confidentiality laws.

## Data Availability

Data are contained within the article.

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
