# Peer review of "A TSHZ3 Frame-Shift Variant Causes Neurodevelopmental and Renal Disorder Consistent with Previously Described Proximal Chromosome 19q13.11 Deletion Syndrome"

_genes, 2022, doi:10.3390/genes13122191_

Round 1

Reviewer 1 Report

René G. Feichtinger etal, performed a case report study. They describe a situation of a 7 year-old girl, with multiple disabilities – intellectual, behavioral or renal- and the connection of them, with a TSHZ3 frame- shift variant. The authors describe how heterozygous deletions at 19q12-q13.11 –inherited from paternal allele-affecting TSHZ3, a gene responsible for encoding a transcription factor important for the appropriate development of neurons and neural system.

They performed a whole exome sequencing technique, using the short-read sequencing platform of illumina, HiSeq 4000, and Sanger sequencing as a validation method. The protocol they applied with the previous sequencing techniques and with SureSelect60Mbv6 (Agilent) for library preparation, Burrows-Wheeler Aligner (BWA, v.0.5.87.5) and PINDEL to detect the breakpoints of deletions, considered to be well design and effective.

We have to point out that, the authors performed adequate literature search, as they mention previous similar cases in the introduction and in discussion, and their clinical description was excellent. We should also highlight that they specified their limitations, namely the inability to receive the paternal DNA.

It is a very interesting study, although in some cases they tend to use strong wording about causation.

One major issue that I have is that there is no description of the rest of the results of the exome sequencing. No other variants were detected? I would need to see at least a summary table as in number of variants detected, synonymous, nonsynonymous etc and the top identified variants.

Lines 20,112,160 typo "allel" to "allele".

Line 40, probably meant "In 1998". Suggest rephrasing.

Line 43, "proves a gene disease association". Strong wording, please ameliorate. I suggest "adds to a growing pool of associations etc". This is also done in some other places as well.

Author Response

Reviewer reply

We thank both reviewers for their useful comments. Please find below a point by point reply to all points raised by the reviewers.

Reviewer 1:

Point 1: One major issue that I have is that there is no description of the rest of the results of the exome sequencing. No other variants were detected? I would need to see at least a summary table as in number of variants detected, synonymous, nonsynonymous etc and the top identified variants.

Reply point 1: Thank you for this important point. We added a table with rare variants (allele frequency less than 6 in 27503 in our database) and another table with genes that have two variants per gene (either bi-allelic or mono-allelic, allele frequency less than 551 in 27503 in our database). These data are intended for the reviewers' use only, since we are not allowed to publish such a huge personal information of our patient. The 20-times coverage was 97.1%, 78832 SNV were detected, 11007 Indels. After filtering, we had 112 rare variants (allele frequency less than 6 in 27503 in our database) and 100 variants with two hits in one gene (either bi-allelic or mono-allelic, allele frequency less than 551 in 27503 in our database). The information of frequencies has been included to the manuscript (line 63-66).

 Point 2: Lines 20,112,160 typo "allel" to "allele".

Reply point 2: We corrected the typo according to the reviewer's suggestion.

Point 3: Line 40, probably meant "In 1998". Suggest rephrasing.

Reply point 3: We corrected the typo according to the reviewer's suggestion.

Point 4: Line 43, "proves a gene disease association". Strong wording, please ameliorate. I suggest "adds to a growing pool of associations etc". This is also done in some other places as well.

Reply point 4: We changed the wording according to the reviewer's suggestion.

Reviewer 2 Report

Feichtinger and colleagues described a new heterozygous frameshift within the TSHZ3 gene and compared their finding with the proximal 19q13.1 deletion syndrome.

In general, the text should be made more precise. There are several typos and errors in punctuation, and a lack of references in the draft.

Major issues:

How did the authors consider the 6 LoF heterozygous in gnomAD database? The presence of heterozygous truncating variations in gnomAD should be discussed.

The authors didn't mention either the heterozygous D242H missense observed by Nicolaou et al in a series of CAKUT patients or the de novo heterozygous stop gain in DECIPHER NC_000019:g.31279427G>T. They should add or discuss these variations.

The authors didn't precise if there were reports of partial deletion of TSHZ3.

As this is the first report of a truncating variation in TSHZ3, the ACMG criteria should be detailed for the classification of the variation. All pieces of evidence used for the classification should be explained (probably not "pathogenic" with ACMG 2015 criteria).

A functional approach with RNAseq (blood, fibroblasts) should be proposed to enforce the description of this new variation.

Minor issues:

in point 2, the methods should be more detailed regarding the restriction sites, the primers...

What database is used for cut-off? Is the "reference entries" (L62) an in-house database?

In point 3.1,

What is the age of milestones?

The authors should use the same nomenclature for the description of the percentiles.

What is the meaning of "a delay in her speech-language" (L79): words? Sentences?

How did the authors explain the difference between the IQ of 77 (borderline) and the performances like a 4 to 5 y/o child compared to the biological age of 7 y/o ?  It is quite discordant.

In point 3.2, why use the pLI, the lof o/e score is used now instead of the pLI which is associated with a lot of bias.

What is the allelic depth for the mother or the proband?

The supplemental figure 1B is unreadable (exom without e).

In the discussion, there are absolutely no references in the text. References to MIM descriptions with MIM numbers are important but insufficient.

Author Response

Reviewer 2:

Point 1: How did the authors consider the 6 LoF heterozygous in gnomAD database? The presence of heterozygous truncating variations in gnomAD should be discussed.

Reply point 1: We thank the reviewer for this point and added the following statement:

"In gnomAD six LoF variants are listed: c.41-1G>T; p.Glu66Ter; p.Gln177Ter; p.Tyr279Ter; p.Met1028AsnfsTer30; p.Pro1068ArgfsTer11; p.Leu1072ProfsTer6. Since in gnomAD also affected individuals are included we had a look at the gnomAD v.2.1.1 (controls ) data set only containing healthy individuals. Here only the p.Met1028AsnfsTer30 and p.Leu1072ProfsTer6 LoF variants are found. The p.Leu1072ProfsTer6 variant is flagged in gnomAD for low confidence (LCpLoF). Only the p.Met1028AsnfsTer30 variant remains in the gnomAD control population. However, this mutation in contrast to the described mutation affects the outermost C-terminus of TSHZ3 and therefore might allow the production of a stable transcript with some residual function."

 Also the p.Leu1072ProfsTer6 variant would affect the C-terminus.

Point 2: The authors didn't mention either the heterozygous D242H missense observed by Nicolaou et al in a series of CAKUT patients or the de novo heterozygous stop gain in DECIPHER NC_000019:g.31279427G>T. They should add or discuss these variations.

Reply point 2: We thank the reviewer for his help and this excellent search. We were not aware of the Nicolaou article and indeed the patient listed in DECIPHER might be a match.

 We included the following two statements:

"In Decipher database (https://www.deciphergenomics.org/), a patient carrying a de novo LoF variant in TSHZ3 c.[366C>A];[=]; p.[Tyr122Ter];[=] and abnormalities of the nervous system is described. Involvement of the nervous system would fit to our individual."

 "A heterozygous missense variant c.[724G>C];[=] p.[Asp242His];[=] was described by Nicolaou et al. in a series of CAKUT patients. Although the renal phenotype would fit to our study it is doubtful that the heterozygous missense variant has a pathogenic relevance (PMID: 26489027). According to the ACMG classification this variant is benign (BP1; BP4; PM2)."

 Point 3: The authors didn't precise if there were reports of partial deletion of TSHZ3.

Reply point 3: We had included two statement that one case was published with a small intragenic deletion of TSHZ3 in line 145/146 and line 162/163.

 Additionally we changed the wording and included the respective reference to highlight this fact a bit more: „A total of 14 individuals with heterozygous deletions encompassing TSHZ3 including one with an intragenic deletion (patient 7 from Caubit et al.) have been described in literature so far (PMID: 27668656).”

 Point 4: As this is the first report of a truncating variation in TSHZ3, the ACMG criteria should be detailed for the classification of the variation. All pieces of evidence used for the classification should be explained (probably not "pathogenic" with ACMG 2015 criteria).

Reply to point 4: The frame-shift c.119_120dup causes by definition a LoF. According to Varsome the TSHZ3 variant of our patient is classified with uncertain significance (PM2). Since the variant is a phylogenetically conserved region of TSHZ3, a pathogenic relevance is likely. The variant is not found in gnomAD. Especially the absence of LoF variants in gnomAD is a very good indicator for importance.

Point 5: A functional approach with RNAseq (blood, fibroblasts) should be proposed to enforce the description of this new variation.

Reply point 5: We included the following statement in line 164/165: "RNA sequencing should be performed to enforce the description of this new variation."

Point 6: in point 2, the methods should be more detailed regarding the restriction sites, the primers...

What database is used for cut-off? Is the "reference entries" (L62) an in-house database?

 Reply point 6: We included detailed information on restriction digestion (site, incubation) as well as PCR. Concerning the database for exome allele frequencies, this is indeed our in-house database with more than 27,000 entries. We added this information to the manuscript.

Point 7: In point 3.1, What is the age of milestones?

Reply point 7: We added the required information to line 83-85

The girl was slightly delayed in achieving early motor milestones (e.g. walking independently from age 24 months onwards, unable to ride a bike but able to swim by the age of 6.5 years), but mainly showed a delay in her receptive and expressive speech and -language (both receptive and expressive), intellectual and social-emotional development.

Point 8: The authors should use the same nomenclature for the description of the percentiles.

 Reply point 8: We corrected the information see line 87

Point 9: What is the meaning of "a delay in her speech-language" (L79): words? Sentences?

 Reply point 9:We corrected the information according to reviewers suggestion see line 85

Point 10: How did the authors explain the difference between the IQ of 77 (borderline) and the performances like a 4 to 5 y/o child compared to the biological age of 7 y/o ?  It is quite discordant.

 Reply point 10: According to the reviewers suggestion we added more detailed results in this section, lines 95- 127

 “At the biological age of 6.9 years she was extensively tested.The Wechsler Intelligence Scale for Children 5th edition (WISC-V), showed a border-line total IQ of 77 (mean 100, SD 15), percentage rank (PR) 6), with the following subscales: Verbal Comprehension Index IQ 100/PR 50; Fluid Reasoning Index IQ 86/PR 18, Working Memory Index IQ79/PR8; Processing Speed Index IQ80/ PR9).

Her speech and language development was tested using German test batteries (e.g. TROG-D for grammar comprehension and PDSS for vocabulary, Vienna test for sentence completion, Scenotest (general developmental projective test in German language) and showed deficits in all areas with a speech age of about 4-5 years.

She had major problems with motor skills (fine motor skills, hand coordination, body coordination, strength and skills, all tested with Bruininks-Oseretzky test; graphomotoric skills tested with the house-tree-person drawing test)) where she showed a developmental age of 4-4.9 years.

performed like a 4-4.9 year-old child and had major problems with graphomotoric and other fine motor skills. In general, it was noticed that she had a lot of problems with everyday tasks, e.g. she was unable to put her socks on or dress herself. Her overweight was attributed to a sensory deficit in body perception. She further had behavioural prob-lems. The Wechsler Intelligence scale for Children 5th edition (WISC-V) showed a total IQ of 77 (mean 100, SD 15), percentage rank (PR) 6), with the following subscales: Verbal Comprehension Index IQ 100/PR 50; Fluid Reasoning Index IQ 86/PR 18, Working Memory Index IQ79/PR8; Processing Speed Index IQ80/ PR9). The Child Behavior Check-list (CBCL/6-18R) showed above average external (T=83) and internal (T = 65) problems and a pathological total value (T=79). Taken together her developmental age was 4.0 -4.9 years at the biological age of 6.9 years.”

Point 11: In point 3.2, why use the pLI, the lof o/e score is used now instead of the pLI which is associated with a lot of bias.

What is the allelic depth for the mother or the proband?

Reply point 11: In addition to the pLI we added the LoF o/e score.

The depth of the TSHZ3 (NM_020856.4) variant c.[119_120dup];[=] (p.[Pro41SerfsTer79];[=]) was 224 (52% of that were the duplication). In the proband, the synonymous variant rs61742321 was covered 117 times (44% variant allele) and the synonymous variant rs12461253 was covered 197 (52% variant allele). All three variants are graphically displayed in Supplementary Figure 1A. The mother was not exome sequenced. However, there was a heterozygous SNP (rs28609894) in her Sanger sequencing results which looked heterozygous, both in forward and reverse direction (data available on request).

The following statement was included in line 110 and 111: “The exome sequencing depth at the TSHZ3 variant c.119_120dup was 224 (52% of that were the duplication).”

 Point 12: The supplemental figure 1B is unreadable (exom without e).

Reply point 12: We corrected this typo.

Point 13: In the discussion, there are absolutely no references in the text. References to MIM descriptions with MIM numbers are important but insufficient.

Reply point 13: We included some references according to the reviewer.

Round 2

Reviewer 2 Report

The presentation of citation in the introduction and the discussion must be harmonised.